# Gut Microbiota and *Clostridium difficile*: What We Know and the New Frontiers

**DOI:** 10.3390/ijms232113323

**Published:** 2022-11-01

**Authors:** Andrea Piccioni, Federico Rosa, Federica Manca, Giulia Pignataro, Christian Zanza, Gabriele Savioli, Marcello Covino, Veronica Ojetti, Antonio Gasbarrini, Francesco Franceschi, Marcello Candelli

**Affiliations:** 1Department of Emergency Medicine, Fondazione Policlinico Universitario A. Gemelli IRCCS, 00168 Rome, Italy; 2Facoltà di Medicina e Chirurgia, Università Cattolica del Sacro Cuore, 00168 Rome, Italy; 3Foundation of Ospedale Alba-Bra, Department of Anesthesia, Critical Care and Emergency Medicine, Michele and Pietro Ferrero Hospital, 12060 Verduno, Italy; 4Emergency Department, Policlinico Universitario San Matteo, IRCCS, 27100 Pavia, Italy

**Keywords:** gut microbiota, *Clostridium difficile* infection, microbiome

## Abstract

Our digestive system, particularly our intestines, harbors a vast amount of microorganisms, whose genetic makeup is referred to as the microbiome. *Clostridium difficile* is a spore-forming Gram-positive bacterium, which can cause an infection whose symptoms range from asymptomatic colonization to fearsome complications such as the onset of toxic megacolon. The relationship between gut microbiota and *C**lostridium difficile* infection has been studied from different perspectives. One of the proposed strategies is to be able to specifically identify which types of microbiota alterations are most at risk for the onset of CDI. In this article, we understood once again how crucial the role of the human microbiota is in health and especially how crucial it becomes, in the case of its alteration, for the individual’s disease. *Clostridium difficile* infection is an emblematic example of how a normal and physiological composition of the human microbiome can play a very important role in immune defense against such a fearsome disease.

## 1. The Gut Microbiota

Our digestive system, particularly our intestines, harbors a vast amount of microorganisms including bacteria, archaea, bacteriophages, eukaryotic viruses, and fungi called the microbiota [1], whose genetic makeup is referred to as the microbiome.

The number of these microorganisms inhabiting the human gastrointestinal tract is extremely high, reaching, according to some estimates, a ratio of equality (about 1:1) vis-à-vis all cells in the human body, while the genetic material of these microorganisms appears to have at least 100 times more genetic diversity than that of the entire human genome [2,3].

A “superorganism” is defined as the set of the host and all the microorganisms that colonize it [2].

In terms of their composition, most of these bacteria belong to the *Firmicutes* phyla (64%), followed by *Bacteroidetes* (23%) and *Proteobacteria* (8%) to Gram-negative bacteria such as *E. coli* and *H. pylori* [4] (Figure 1).

The various gastrointestinal regions, because of their different characteristics, represent different microenvironments, in which specific microorganisms grow [5].

The stomach has special characteristics, such as its acidic ph, so most of the microorganisms that colonize it are acid resistant.

The most important microorganism residing in the gastric lumen is Helicobacter pylori, which influences the growth of other secondary species, which may play a mutualistic or pathogenic role [6].

Regarding the small intestine, it is characterized by the presence of oxygen, rapid luminal flow, and bactericidal secretions such as bile acids.

In the duodenum, the predominant phyla are Firmicutes and Actinobacteria [7].

The colon, on the other hand, is in a condition of anaerobiosis, where the slower passage of food, absorption of water, and fermentation of undigested food take place.

For these reasons, the most common microorganisms are *Bacteroides*, *Bifidobacterium*, *Streptococcus*, *Enterobacteriaceae*, *Enterococcus*, *Clostridium*, *Lactobacillus*, and *Ruminococcus* [8] (Table 1).

This type of complex mutualistic interaction between these microorganisms and their host appears to have evolved over thousands of years [2].

In fact, the ingestion of germs that will later go on to contribute to the formation of human intestinal flora has been known since ancestral times, traces of which have been found in cave paintings, dating as far back as the Neolithic age [3].

The interaction between diet, microbiota, and the human host has fascinated researchers since the turn of the last century when Metchnikoff tried to link this complex interaction and senescence [4].

One of the controversial aspects that characterize this topic, is that in the early days, it was not very clear whether it was the microbiota that predisposed toward certain pathological conditions or whether the opposite was the case [9], while currently, most researchers argue that it is an alteration of the gut microbiota that predisposes toward the onset of certain diseases.

Most of these microorganisms play a commensal role with their host [9].

In fact, this complex host–individual interaction could become so long-lived only by bringing, most of the time, benefits to both parties.

In recent years, more and more efforts are being made to investigate certain aspects that characterize the microbiome, and one of these is to understand its interindividual diversity.

The mechanism by which each individual develops his or her own microbiome appears to be multifactorial in origin, with several factors being called into play, some genetic and others environmental such as those related to childbirth and early life, followed by the type of infant feeding, medications taken, and lifestyle [9].

We have seen how the gut microbiota is greatly affected by environmental influences.

An interesting line of research is on strategies to maintain good health through the homeostasis of the gut microbiota.

Some authors suggest practicing a vegetarian diet, while wheat gluten, red meat, and alcohol are related to dysbiosis that can trigger a chronic inflammatory response [10].

The very fact that so many factors come into play therefore makes it very arduous to know about each one.

Another fascinating field of study in recent years is trying to identify and understand the many functions performed by the microbiome.

The roles played by the gut microbiota are multiple, going on to act on food digestion, drug metabolism, and regulation of intestinal endocrine function, and going as far as being involved in host immune defense mechanisms (Table 2) [1].

The roles played by the microbiota are indeed multifaceted; in fact, it is not surprising to learn that some researchers are focusing on the possible role of microorganisms in fighting cancer, a new frontier with little evidence at the moment but very promising [11].

Again, the real challenge is to try to disentangle all these different functions that are performed in different individuals.

In our case, we will see how crucial its protective function is in the healthy individual and how disastrous its lack is in patients who will develop *Clostridium difficile* infection.

## 2. *Clostridium difficile*

*Clostridium difficile* is a spore-forming Gram-positive bacterium, which can cause an infection whose symptoms range from asymptomatic colonization to fearsome complications such as the onset of toxic megacolon [12].

It is this broad spectrum of clinical manifestations that makes this disease so insidious.

Detection of *C. difficile* without evidence of clinical signs is termed colonization, while *Clostridium difficile* infection (CDI) is referred to when *C. difficile* is present associated with its characteristic clinical manifestations [13].

CDI is a clinical condition that doctors in different specialties very often struggle with.

CDI is one of the most common nosocomial infections [8,14].

By some estimates, CDI affects about 460,000 people a year in the US [15].

These important data help us understand why this disease is increasingly proving to be one of the big public health topics.

The main risk factors for CDI appear to be older age (above 65 years), antibiotic use, and nosocomial exposure [16].

Several considerations can be drawn from these data.

One is purely epidemiological: with the inevitable increase in the geriatric population, these numbers are unfortunately bound to increase.

Another, however, is that all three of the conditions mentioned are acquired.

It is precisely these observations that underlie the involvement of the microbiome in the pathogenesis of this disease.

The onset of *CDI* occurs as a result of the oro-fecal transmission of sufficient numbers of spores of a toxin-producing strain of *C. difficile* within the colon of the host, accompanied by their overgrowth at the expense of normal commensal microorganisms [17].

The occurrence of *Clostridium difficile* infection is thus associated with an alteration in the gut microbiota by antibiotics that are not active against *C. difficile*, which thus cause uncontrolled growth [18].

A great many antibiotics are therefore associated with the occurrence of *CDI*, and those with a higher risk appear to be more commonly used antibiotics including penicillins such as amoxicillin and ampicillin, as well as cephalosporins, clindamycin, and fluoroquinolones [14]. Thus, the main risk factor for *CDI* appears to be exposure to broad-spectrum antibiotics, precisely because of the changes they make to the normal human microbiome [19] (Figure 2). Asymptomatic colonization on admission to the hospital is estimated to have a prevalence of 0.6–13% [20].

These data seem to explain such a high incidence of *CDI*, the genesis of which is multifactorial.

Thus, the factors at play are asymptomatic colonization, associated with antibiotic therapy that brings an imbalance to the microbiome, which may then promote the onset of CDI.

This is why CDI is often considered the best example of the symbiotic relationship between the individual and the host and how its imbalance can lead to disastrous consequences.

*C. difficile* is rarely invasive, and its mechanism of intestinal damage appears to be mediated by its potent exotoxins: toxin A and toxin B [21].

Treatment of CDI therefore relies on the use of antibiotics such as fidaxomicin, vancomycin, and metronidazole [22].

Fidaxomicin has proven to be the most selective antibiotic against *C. difficile* [23] and has been shown to be a highly effective treatment against this disease, although more data are needed regarding safety and efficacy in children and adults [24].

Vancomycin appears to be less selective than fidaxomicin against other intestinal bacteria [23] and continues to be the cornerstone treatment of CDI along with fidaxomicin.

Previously, metronidazole was considered along with vancomycin to be the pivotal treatment for CDI, until new guidelines in 2017 considered it to be of lower efficacy than fidaxomicin and vancomycin, which thus became the main antibiotics for the treatment of CDI [25].

Prolonged use of metronidazole is also burdened by important side effects such as neurotoxicity [26].

As antibiotic treatment is burdened with an important risk of relapse, new therapeutic strategies to cope with CDI are under investigation, which we will discuss later [25].

It should also not be forgotten that for very severe patients, those with a “fulminant” form of *C.m difficile* colitis who present with major symptoms such as shock, hypotension, and megacolon, surgical therapy is also considered [27].

Precisely because *CDI* is such a complex disease, it can require such diverse treatments.

However, as we shall see, even when a *CDI* is appropriately treated with the use of antibiotics, the serious problem of recurrence can arise, which is particularly significant for this disease.

The risk of recurrence remains significant, ranging from 12 to 64 percent, with a median of 22 percent [28].

Recurrence is said to occur when, after treatment accompanied by a total disappearance of clinical symptoms, there is a recurrence of CDI within two to eight weeks after discontinuing therapy [27]. 

Treatment of some of these forms of this recurrent colitis relies on the use of fecal microbiota transplantation (FMT), which is the installation of treated feces collected from a healthy donor into a patient with CDI [29,30].

Regarding FMT, there are some aspects that differ in different centers, such as the most effective dose and preparation time, while some aspects are absolutely shared, such as donor exclusion criteria.

In fact, one of the crucial aspects of this treatment is the careful selection of the donor, as there is a potential risk of pathogens through the FMT procedure.

Thus, patients who carry or are at risk of transmitting infectious diseases, those with gastrointestinal diseases, or who may have recently taken certain medications (antibiotics, immunosuppressants, etc.) that alter the composition of the gut microbiota are excluded [31].

Regarding the safety of this procedure, some data show that adverse effects occurred in 9.2% of patients, including death (3.5%), infection (2.5%), and recurrence of intestinal bowel disease (0.6%) [32].

There are many routes of administration of FMT: oral capsules, procedures involving the upper gastrointestinal tract such as the nasojejunal and nasoduodenal tube, and those of the lower gastrointestinal tract such as colonoscopy or enema.

The lower route of administration appears to be more effective than the upper route, although further studies are needed in this regard [33]. 

The use of FMT has also been considered for the treatment of fulminant colitis supported by several pieces of evidence, including a retrospective study of 199 patients, in which FMT was associated with reduced mortality and colectomy rates [34].

Meanwhile, evidence on the use of FMT for the treatment of recurrent CDI comes from both randomized trials [35] and meta-analyses [36].

FMT represents one of the latest frontiers of CDI treatment and is an ever-evolving field, in which the role of the microbiome is of paramount importance (Figure 3).

## 3. Interactions between the Gut Microbial Communities and *Clostridium difficile* Infection

### 3.1. What We Know

The relationship between gut microbiota and *C**lostridium difficile* infection has been studied from different perspectives (Table 3).

*C. difficile* grows in the digestive tract of infants, where it does not develop an infection but rather a colonization, until during growth when new microbial species take over, providing protection against *C*. *difficile* itself, although any new imbalance may again stimulate its growth [37].

The composition of the gut microbiota then, after a rapid change in early life, remains nearly stable throughout adulthood, until undergoing new changes with advancing age.

Colonization resistance refers to the peculiar ability on the part of the gut microbiome to resist colonization against pathogenic organisms, including *C*. *difficile*.

Several mechanisms underlying this resistance to colonization against *C*. *difficile* have been hypothesized, ranging from stimulation of host immune defenses, competition for nutrients, production of a protective physical barrier against the intestinal mucosa to production of inhibitory substances such as secondary bile acids and bacteriocins [38].

### 3.2. Healthy Carriers

Asymptomatic colonization of *C. difficile* without any signs of disease has been described in many studies on both human beings and animals [39].

The literature suggests that the range of intestinal *C. difficile* colonization of healthy adults can rate from 2.4% to 17.5% [40], and it can be associated with the composition of intestinal microbial communities but also with other extrinsic factors, such as the living environment as well as the host immune state. 

Ozaki et al. reported that *C. difficile* colonization was relatively common among healthy individuals. A healthy asymptomatic carrier has no major difference regarding the gut microbial community compared to a healthy subject. Therefore, Rea MC et al. suggested that the commensal flora in such subjects could protect the host by preventing potentially pathogenic *C. difficile* colonization, multiplication, and toxin production [41].

However, unlike a healthy person with a negative culture, the carrier subject can develop CDI because of the changes in the microbiome that may contribute and favor the growth of the microbe and eventually CDI such as antibiotic treatment, diet, age, host immune state, environment, and hospital admission. 

The presence of *C. difficile* in healthy carriers is one of the most important pieces of evidence of one of the many roles played by normal commensal flora.

In this case, the microbiome performs its physiological and essential immune regulatory function, thus succeeding in curbing even a dreaded condition such as *CDI*.

### 3.3. Colonization in Infants

In fecal samples from newborns and infants, the presence of *C. difficile* rates around 70% [42].

Environment, hospitalization, and prematurity have been associated with the colonization of the bacterium. Some studies showed that vaginal delivery and maternal genital tract are not a risk for newborn acquisition. Vaginal swabs of mothers just before delivery have been examined, and they were all negative for *C. difficile* by culture, but their infants were positive.

Penders et al. [43] associated increased colonization with *C. difficile* with birth by cesarean delivery. The newborns had lower numbers of Bifidobacterium and Bacteroidetes and were more often colonized with *C. difficile* compared with vaginal delivery infants.

Rousseau et al. observed that some microbial taxa such as *Bifidobacterium longum* were protective and negatively correlated to *C. difficile* colonization. In contrast, *Ruminococcus gravus* and *Klebsiella pneumoniae* were susceptible microbial taxa [44].

Plus, the feeding method used has been reported as a factor that can influence *C. difficile* colonization. Breast milk is a more protective factor than formula milk. That can be explained, as some studies showed, by an increase in the acidity in the intestine contents, which may facilitate sporulation and reduce vegetative forms [45].

All of this evidence only reiterates how the gut microbiota begins to form and perform its functions from the earliest days of life, and again, there are multiple factors involved in the development and regulation of the microbiome.

Proteins of human milk play an inhibitory role in toxin TcdA [46].

Moreover, secretory IgA has shown neutralizing activity against toxin A [47].

On the other hand, formula-fed infants are often colonized with *Escherichia coli*, *C. difficile*, Bacteroides, and Lactobacilli compared with breast-fed infants [43].

What is important to underline is that toxigenic and non-toxigenic *C. difficile* has been isolated in infants with a higher percentage of the non-toxigenic one. However, even with the presence of toxigenic strains, colonization seems not to be associated with *CDI*. 

So, the infant gut appears to be resistant to CD toxins. That could be explained by the absence of toxin receptors, poorly developed cellular signaling pathways because of the immature gut mucosa, or the presence of protective factors in the infantile gut that remains at the moment unknown [47].

As we have just seen, this fundamental interaction plays an essential and delicate role from an early age.

## 4. *Clostridium difficile* Infection

### 4.1. Disruption of the Microbiome and Clostridium difficile Infection Risk Factors

Clearly, the most well-known risk factor for developing *CDI* is antibiotic use, both short- and long-term because of its impact on microbiota diversity. In fact, a healthy microbial community gut is capable of interfering with *C. difficile* spores, and it does not necessarily result in disease. However, every change in the microbial environment could bring spore germination, CD growth, and toxin production [48].

Another known risk factor is increasing age. In elderly people, the structure of the microbiome undergoes changes, is less diverse, and shows a decrease in protective species such as *Bifidobacteria* and some *Firmicutes*, as well as an increase in *Bacteroidetes* and *Proteobacteria* [49]. 

That change also influences the immune system of an elderly person which becomes weaker and more fragile. As a matter of fact, the rate of *C. difficile* infection is higher for people aged 65. Advancing age is also associated with hospitalization, use of antibiotics, and development of different diseases [50,51,52,53,54].

Another risk factor is proton pump inhibitors (PPIs), which increase the gastric pH and modulate microbiota, influencing above all *Lactobacillus* [51].

Gastrointestinal pathologies such as inflammatory bowel disease (IBD) may impact *C. difficile* susceptibility. As some studies show, we observed in subjects suffering from IBD a decreased diversity of *Firmicutes* and *Bacteroidetes* and the presence of pathogenic bacteria such as the *Proteobacteria* phylum [52]. 

Moreover, its inflammatory products in IBD (antimicrobial peptides lipocalin-2 and calprotectin) potentially impact the growth of surrounding microbes [53].

Differently, as a protective factor, we can safely say that a good composition of microbial communities’ gut and a strong immune response could be useful against CDI. 

Serum IgG antibodies against toxins A and B have been associated with protection in some studies [54].

So immunization, both active and passive, could be a good strategy to study for CDI treatment [55].

Therefore, a full understanding of all these risk factors is of paramount importance for the development of new preventive strategies to avoid the occurrence of this dangerous infection.

### 4.2. Recurrent Clostridium difficile Infection and the Incomplete Recovery of the Microbiota

The most common complication of *CDI* is incomplete recovery and recurrent infection. The rates are about 20.30% after an initial infection and up to 60% after three infections [56]. 

Some studies notice that each episode of CDI brings an increasing chance of recurrence, also due to an increase in antibiotic use and disease severity [57].

However, it is also hypothesized that antibiotic treatment interferes with the ability of the gut microbiota to recover fully and re-establish colonization resistance in some individuals. Alternatively, recurrence could reflect the failure of the host to mount a protective immune response against *C. difficile* [58].

This is how an imbalance, an acquired rupture of this delicate balance between the host and the individual, can degenerate into an almost irreparable situation.

We can safely assert that being able to prevent one of the most frequent complications, namely the recurrence of this widespread infection, is one of the great challenges of public health.

### 4.3. Fecal Microbiota Transplantation in Clostridium difficile Infection

More recent work has studied fecal microbiota transplantation (FMT) for recurrent *Clostridium difficile* infection.

Patients with severe CDI refractory to traditional antibiotic treatment have had success with FMT, which restores colon homeostasis by reintroducing bacteria from healthy donor stool. The success rate for FMT is greater than 90% for those who had recurrent CDI, but the mechanism behind this treatment is partially unknown [59].

Successful fecal microbiota transplantation is correlated with a dysbiosis resolution by the replenishment of *Roseburia* and *Bacteroidetes*, which are also involved in butyrate production. After fecal transplantation, studies reported the presence of an increase in richness and diversity, an eradication of *Proteobacteria* species, and a restoration of *Firmicutes* and *Bacteroidetes* species [60].

They also found that the patients’ gut communities were completely restored within three days following fecal transplantation, with stability in species for at least four months and indistinguishable from that of the donor [61].

Following FMT, *Bacteroidetes* increased and *Proteobacteria* decreased. Some others observed protective microbial taxa are *Alistipes, Ruminococcaceae, Lachnospiraceae, Peptostreptococcaceae,* and *Verrucomicrobiaceae*. All the species are negatively correlated to *C. difficile* colonization [62] (Figure 4).

This is an emblematic example of how by being able to re-establish that fundamental relationship between the host and gut microbiota, one of its most important functions, namely that of immune regulation, can be recovered.

All these new discoveries undoubtedly make it one of the most promising and steadily growing fields in all of medicine.

## 5. New Therapeutic Strategies

With this close link between the microbiome and the onset of *Clostridium difficile* infection, several lines of research are currently underway aimed at developing new therapeutic strategies (Table 4).

Despite all this new evidence, there is currently still not full unanimity on defining the microbial species associated with asymptomatic colonization versus those related to the onset of CDI, which remains one of the current challenges to be addressed [47,63].

One of the proposed strategies is to be able to specifically identify which types of microbiota alterations are most at risk for the onset of *CDI* [38].

This field is particularly promising due to the fact that we are increasingly witnessing significant technological development, with reduced sequencing costs.

In this case, it is the evolution of science in a broad sense, understood as the development of new technologies that are increasingly within the reach of researchers that could also prove decisive in this specific area of study.

Other authors, on the other hand, suggest how diet may influence the composition of the microbiota and consequently its interaction with various pathological conditions, including CDI, thus looking for a correlation between diet and the development of this disease [64].

This aspect also appears to be much studied for other diseases such as metabolic syndrome and chronic inflammatory bowel disease, less so for CDI.

From the pathophysiological point of view, the connection between diet and the microbiome is very clear.

However, the occurrence of *CDI*, as we have seen, also calls into question several other factors, and that is why it makes this field of study very difficult, albeit very promising.

Meanwhile, new therapeutic frontiers make use of probiotics and prebiotics.

Probiotics are “Live microorganisms that, when administered in adequate amounts, confer a health benefit upon the host” [65], while prebiotics are “A selectively fermented ingredient that allows specific changes, both in the composition and/or activity in the gastrointestinal microflora that confers benefits upon host well-bring and health” [66].

Meanwhile, “synbiotic” means the administration of a prebiotic together with a specific probiotic to enhance the engraftment and development of that specific microbe [67].

A study was recently conducted in which treatment with live purified Firmicutes bacterial spores was successfully proposed for patients with recurrent CDI [68].

An increase in the concentration of secondary bile acids compared to primary bile acids is one of the factors inhibiting the germination of CD spores.

For this very reason, spore-forming *Firmicutes* bacteria were grafted with the specific purpose of increasing the concentration of secondary bile acids to thus inhibit spore germination and subsequent CD growth [68].

Here, we have an important finding of how an accurate understanding of this complex individual–host interaction can provide us with important new therapeutic weapons.

Some of the latest research involves in vitro experiments and makes use of synbiotics, bacterial secreted compounds that inhibit the activity of *C. difficile* toxins [67].

These research studies, though currently relegated to the early stages, appear to be very promising.

Live recombinant biotherapeutic products (LBPs) are defined as microorganisms that have been genetically modified by the targeted addition, deletion, or modification of genetic material [69].

Several clinical trials on CDI are currently underway: one is biotherapeutic, whose potential use in CDI relapses is being studied [70], while another one involves the use of Gram-positive selective-spectrum antimicrobials [71] for *CDI* patients.

Again, efforts are being made to counteract this disease by looking at the factors that regulate host immunity.

Other researchers have focused on studying the interaction between the onset of CDI and the presence of valerate, a short-chain fatty acid produced by amino acid fermentation in the microbiota.

Valerate appears to be one of the factors that inhibit the growth of *C. difficile*; in fact, its levels increase after fecal microbiota transplantation [72].

Some of the most surprising research involves the use of bacteriophages.

Bacteriophages are viruses that can go to specific targets, going on to infect and replicate within the designated host [69].

Their potential and undisputed strength would be to go against *C. difficile* without the use of antibiotics [3,69].

There has also been speculation that this particular therapeutic weapon could be used against *C.*
*difficile*, but without obtaining important evidence at present.

A very promising paper was recently published, in which ADS024, a newly characterized strain of *Bacillus velezensis*, was identified that appears to have strong activity against *C. difficile* with negligible impact on the rest of the bacterial flora, and also having protease activity directed against TcdA and TcdB toxins which we have seen to be responsible for the clinical manifestations of this disease [73].

We fully share the enthusiasm shown by Khanna [74] in a recent article in which he summarized the four ongoing trials of capsule therapies (CP101, RBX7455, SER-109, VE303) plus the one based on the use of enema (RBX2660), again related to remodeling the microbiota to cope with *CDI*.

In conclusion, we also point out an excellent paper by Rusha et al. [75] who expounded an excellent summary of probiotic strains used in human clinical trials to treat CDAD.

**Table 4 ijms-23-13323-t004:** New strategies for the prevention of *Clostridium difficile* infection through its interaction with the gut microbiota.

New Therapeutic Strategies	
Aim in the future for increasingly accurate identification of microbiota alterations responsible for the onset of CDI	(Revolinski et al., 2018) [38]
Investigating the relationship between diet, microbiome, and the development of CDI	(Shaji et al., 2022) [64]
Oral administration of pore-forming *Firmicutes* bacteria to prevent recurrence of CDI (phase 3, double-blind, randomized, placebo-controlled study of 182 patients, with a safety profile similar to placebo; has superior efficacy for prevention of recurrent infections)	(Feuerstadt et al., 2022) [68]
Use of symbionts (in vitro studies)	(Mills et al., 2018) [67]
Biotherapeutic and Gram-positive selective-spectrum antimicrobials (clinical trials in progress)	(Orenstein et al., Garey et al., 2022) [70,71]
Identify products of the human microbiota that counteract the occurrence of *Clostridium difficile* infection	(McDonald et al., 2018) [72]
Counteracting *Clostridium difficile* without the use of antibiotics by using bacteriophages	(Zhang et al., 2022) [69]
Identification of ADS024, a new potential therapeutic bacterium directed against *Clostridium difficile*	(O’Donnell, et al. 2022) [73]

## 6. Conclusions

In this article, we carefully explored aspects concerning the complex interaction between *C. difficile* and gut microbiota.

The microbiome is a complex system whose fundamental importance we understand more and more, both for the health and disease of the individual.

It will still take a long time to fully understand it, for multiple reasons, including its extreme variability in different individuals and its involvement in so many different functions.

*C. difficile* infection is one of the great public health challenges.

It is one of the most frequent infections, where the role of its interaction with the microbiota is crucial.

From these assumptions, several therapeutic strategies to deal with CDI have arisen and are being studied.

The most intuitive one is to go after the microorganisms themselves that cause this disease, as is currently done with the use of antibiotics, or by experimenting with new and different innovative methods, such as bacteriophages.

The other strategy, on the other hand, is to work on the substrate that promotes the pathogenesis of *CDI*.

In this regard, we mentioned fecal microbiota transplantation and all the other numerous therapeutic strategies that are still being researched or tested.

We also saw how it is not enough to eliminate the bacteria that cause CDI since in the absence of a protective substrate from the microbiome, the risk of recurrence is extremely high.

It is precisely in order to prevent these recurrences that the use of fecal transplantation, one of the latest therapeutic frontiers involving the gut microbiota, is being employed.

As we discussed extensively, research on these issues is more alive than ever, and we hope it will continue to be so that we can have new therapeutic weapons against this infection.

## Figures and Tables

**Figure 1 ijms-23-13323-f001:**
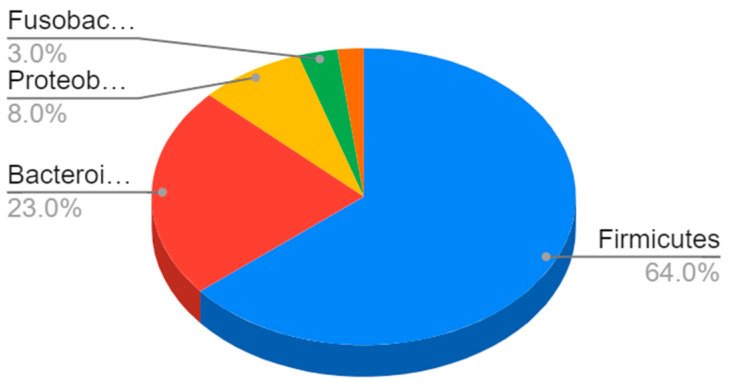
Composition of the gut microbiota. The most represented phyla are Firmicutes (64%) and Bacteroidetes (23%), followed by Proteobacteria (8%), Fusobacteria, verrucomicrobia, and actinobacteria (3%).

**Figure 2 ijms-23-13323-f002:**
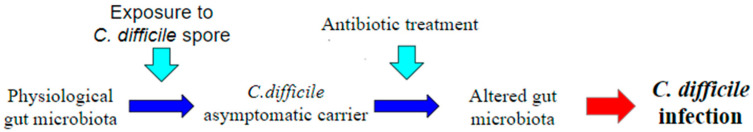
Pathogenesis of *Clostridium difficile* infection (CDI). One of the most accepted theories to explain the onset of CDI is the administration of antibiotics in a healthy *C. difficile carrier* patient. Antibiotic treatment, by generating a dysbiosis in the gut microbiota, causes a loss of its defense function, leading to the onset of CDI.

**Figure 3 ijms-23-13323-f003:**
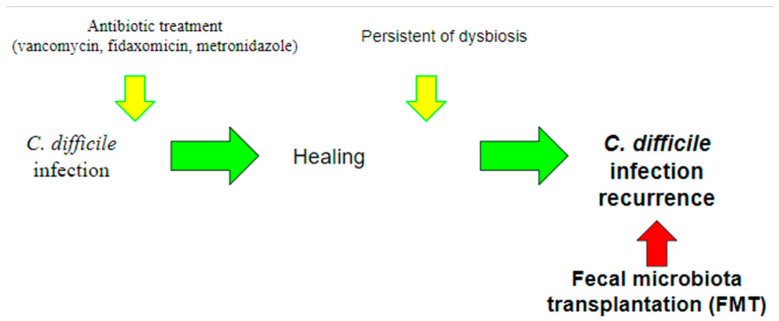
Use of fecal microbiota transplantation (FMT). Currently, the use of FMT is indicated for the treatment of recurrent CDI, in which recurrent infections occur due to the persistence of dysbiosis. The aim of this treatment is precisely to intervene by restoring the balance of the gut microbiota.

**Figure 4 ijms-23-13323-f004:**
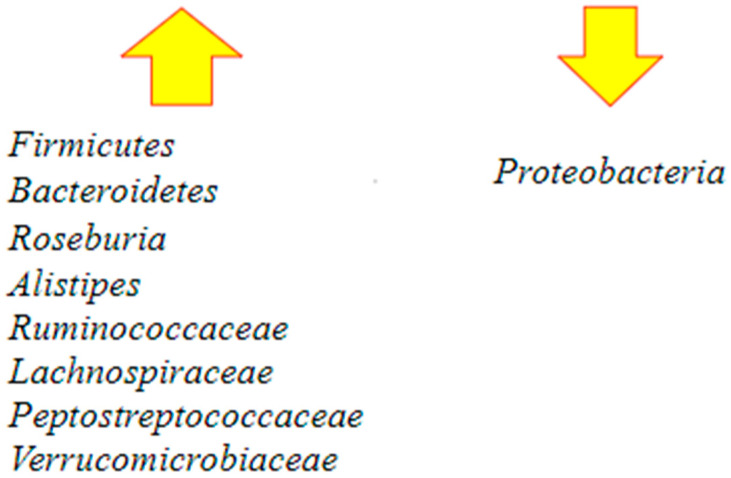
Major changes in the gut microbiota following fecal microbiota transplantation (FMT) [60,62]. After the procedure, there is an increase in the richness and diversity of the microbiome and the restoration of *Bacteroidetes* and *Roseburia*, which are involved in butyrate production. There is also restoration of *Firmicutes* and other protective microbial taxa such as *Alistipes*, *Ruminococcaceae*, *Lachnospiraceae*, *Peptostreptococcaceae*, and *Verrucomicrobiaceae*. At the same time, an important decrease in Proteobacteria occurs.

**Table 1 ijms-23-13323-t001:** Summary of the predominant microorganisms in the various tracts of the digestive system with their basic characteristics.

Gastrointestinal Tract	Characteristics	Predominant Microorganisms
Stomach	Acidic environment	*Helicobacter pylori*
Small intestine	Plenty of oxygen, secretion of bactericidal substances, and rapid luminal flow	*Firmicutes* and *Actinobacteria*
Colon	Slow transit of food, anaerobic condition, site of water absorption, and fermentation of undigested food	*Bacteroides*, *Bifidobacterium*, *Streptococcus*, *Enterobacteriaceae*, *Enterococcus*, *Clostridium*, *Lactobacillus*, and *Ruminococcus*

**Table 2 ijms-23-13323-t002:** Gut microbiota functions.

Gut Microbiota Functions	
Influences	Development and function of the immune system, bone density, pathogen growth, gut endocrine functions, neurologic signaling.
Biosynthesis	Vitamins, steroid hormones, neurotransmitters
Metabolism	Drugs, xenobiotics, bile salts, food components, amino acids

**Table 3 ijms-23-13323-t003:** Summary of some existing interactions between microbiome and the occurrence of *Clostridium difficile* infection.

Relationship between Gut Microbiota and *Clostridium difficile* Infection
**Healthy carriers**—Up to 17.5% of adults are healthy carriers of *Clostridium difficile*, who do not develop the disease protected by commensal bacterial flora.
**Colonization in infants**—In fecal samples from newborns and infants, the presence of *Clostridium difficile* rates around 70%; the infant gut appears to be resistant to *Clostridium difficile* toxins.
**Disruption of the microbiome and CDI risk factors**—Alterations in the microbiota can lead to the onset of CDI. Risk factors that can lead to this include antibiotic use, age, PPI use, and presence of IBD, while having been affected by CDI is a serious risk factor for recurrence.
**Fecal microbiota transplantation in CDI**—Used for some relapsed forms, precisely because this therapy aims to resolve the dysbiosis that led to the onset of the infection.

## Data Availability

Not applicable.

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
