# Peer review of "Gut Microbiota and Clostridium difficile: What We Know and the New Frontiers"

_ijms, 2022, doi:10.3390/ijms232113323_

Round 1

Reviewer 1 Report

The manuscript by Piccioni and collaborators is very interesting and concise. However, as a review which approaches the several aspects of c. dificile, figures and diagrams to facilitate the content are very important. I therefore, recommend the authors to illustrate sections to improve the MS.

Reviewer 2 Report

The topic of review articles is fitted well in scope of journal. Articles is focused on the C. difficile infections and role of gut microbiome. Authors nicely gathered the information on all aspects, combined and drafted. When we think of review articles, we expect an overview, a summary, and an evaluation (critique) of the current state of knowledge, a discussion of methodological issues and suggestions for future research. This manuscript lacks critical evaluation of each information added and needs lot improvement in certain areas.

Major Comments

1.       No line numbers in manuscript, which make life harder to review and comment specifically. No set paragraph, information missing link in between the text.

2.       This manuscript lacks CDI treatments info. Not much discussion about current therapy (Fidaxomicin/Metronidazole). Please talk about it. Fidaxomicin is excellent antibiotic which selectively targets C. difficile which was required when you are dealing with the gut infection. But there is no data supports the safety and effectiveness in children and Adult. Metronidazole is used but prolonged use resulted into neurotoxicity. Vancomycin generally safe but also targets other bacteria in gut. So, we have no full proof treatment for CDI, highlight this fact and talk about new antibiotics in pipeline with their clinical data.   

3.       FMT is promising, but many unknowns exist. Please talk about most effective dose, mtd. Of preparation, route of administration, concern regarding donor sample and clinical trials data.

Minor comments

1.       Page 1, Abstract, please mention Clostridium difficile once and later as C. difficile. Be consistent with it throughout the manuscript and same for other bacteria as well.

2.       Page 1,: Please mention Clostridium difficile infection once and use abbreviate form later “CDI”.

3.       Page 1, line quoting ref 4: Please change gram to Gram here and elsewhere.

4.       Page 2, line quoting ref 5: Please change mychrobioma to “microbioma”.

5.       Page 4, line quoting after ref 21: Why gut is in capital here, be consistent.

6.       Page 5, Check the spelling, - Klebsiella pnumoniae.

7.       Page 5, why Difficile is in capital, lactobacilli in lower case. Please check thoroughly these mistakes before sending MS to publication house.

8.       Page 5, please change toxic to toxin

9.       Page 9, Conclusion, please change Microbiome is a complex…. “organism” to “system”.

Reviewer 3 Report

The manuscript by Piccioni et al. “Gut microbiota and clostridium difficile: what we know and the new frontiers” is very weak and lacks novelty or significance. The manuscript can be improved and re-submitted.

Comments

1.      Overall, the manuscript can be more precisely presented. Please avoid the use of too many paragraphs in each section. Please provide 3-4 paragraphs only in each section of the manuscript. The organism's name (in italic) should be cross-varified in the text (first full name (i.e. Clostridium difficile) followed by the abbreviation (i.e. C. difficile).

2.      Abstract, please delete the first sentence as typical to the introduction. It can be in a single paragraph.

3.      Please add the first section as an Introduction with detailed information about microbiota, microbial role in the community, its importance, distribution of microbiota in guts parts, the implication of perturbation of microbiota, and strategy to maintain microbiota for good health, and finally, objective and significance of this article with recent citations i.e.  Science of The Total Environment (2022) 834:155300;  Indian Journal of Microbiology (2020) 60:420-429. Also, the importance of Clostridium in health i.e. Seminars in Cancer Biology (2021) doi:10.1016/j.semcancer.2021.05.012.

4.      Authors should add some illustrations (2-3) about microbial diversity and population dominance in the gut, factors influencing gut microbiota with descriptions, and strategies to maintain the gut microbiota for better health.  

5.      Tables should be more elaborated and briefly presented with additional entries. Please add more quantitative data, gut microbial population details, location in the gut,  and a brief discussion appropriately in the corresponding sections. 

Round 2

Reviewer 1 Report

As suggested, the authors have now included tables and figures in order to better summarize the information discussed in the review. 

Author Response

We did it

Please see the previous attachment

Reviewer 2 Report

Now the MS looks more promising, after addressing all the suggestion of reviewers.

Reviewer 3 Report

accept

Author Response

We did it

Plese see the previous attachement